# Prolonged bacterial lag time results in small colony variants that represent a sub-population of persisters

Clément Vulin[1,2], Nadja Leimer[3,4], Markus Huemer [3], Martin Ackermann[1,2] & Annelies S. Zinkernagel[3]

Persisters are a subpopulation of bacteria that are not killed by antibiotics even though they lack genetic resistance. Here we provide evidence that persisters can manifest as small colony variants (SCVs) in clinical infections. We analyze growth kinetics of *Staphylococcus aureus* sampled from in vivo conditions and in vitro stress conditions that mimic growth in host compartments. We report that SCVs arise as a result of a long lag time, and that this phenotype emerges de novo during the growth phase in various stress conditions including abscesses and acidic media. We further observe that long lag time correlates with antibiotic usage. These observations suggest that treatment strategies should be carefully tailored to address bacterial persisters in clinics.

[1] Institute of Biogeochemistry and Pollutant Dynamics, ETH Zurich, Zurich 8092, Switzerland. [2] Department of Environmental Microbiology, Eawag, Dubendorf 8600, Switzerland. [3] Division of Infectious Diseases and Hospital Epidemiology, University Hospital Zurich, University of Zurich, Zurich 8091, Switzerland. [4] Present address: Antimicrobial Discovery Center, Department of Biology, Northeastern University, Boston, MA 02115, USA. These authors contributed equally: Clément Vulin, Nadja Leimer, Markus Huemer. These authors equally supervised: Martin Ackermann, Annelies S. Zinkernagel. Correspondence and requests for materials should be addressed to M.A. (email: martin.ackermann@env.ethz.ch) or to A.S.Z. (email: annelies.zinkernagel@usz.ch)

Bacterial infections continue to be a major health threat despite the availability of antibiotics. Antibiotic treatment often fails because bacteria are genetically resistant to antibiotics. However, even among genetically susceptible bacteria, there are phenotypic subpopulations that tolerate antibiotics and survive treatment. These subpopulations, often referred to as persisters[1–3], are growth arrested bacteria, and the growth arrest confers tolerance towards antibiotics. In clinics these persisters may resume growth and result in recurrent bacterial infections. Thus, by alternating between non-growing and growing states fully antibiotic susceptible bacteria may cause chronic and recurrent infections despite the use of antibiotics that are effective in vitro.

Persisters are found in various bacterial species including *Staphylococcus aureus*, *Mycobacterium tuberculosis*, *Escherichia coli*, *Salmonella* Typhimurium, and *Pseudomonas ssp.* and are understood to be linked to clinical persistence[4]. We focused our work on *S. aureus*, one of the major causes of both community-associated and health care-associated infections. This Gram-positive bacterium is of concern due to its propensity to cause chronic and relapsing infections including osteomyelitis, endocarditis, and recurrent abscesses[5–8]. In addition, *S. aureus* infections can relapse after long asymptomatic periods. When sampled from various body sites such as bones and joints, the respiratory and urinary tracts[9,10] and grown on solid agar plates, *S. aureus* sometimes shows marked variation in colony sizes: i.e., some colonies are much smaller than most others from the same sample. These small colonies are referred to as small colony variants (SCVs). More precisely, SCVs are defined as colonies whose size is 5 or 10 times smaller than the most common colony type[11].

Most SCVs recovered in vitro and from murine infection models revert to a normal sized colony upon sub-cultivation and are consequently termed nonstable SCVs[12]. Within a population, these nonstable SCVs usually occur at a low frequency, which increases after exposure to the host intracellular milieu or after exposure to distinct stress conditions, such as acidic pH[13–18] (for *Salmonella enterica*, see ref. [16]). Due to their low frequency within a population, detection and subsequent characterization of nonstable SCVs is challenging and their growth dynamics have not been studied so far. Furthermore, due to their propensity to revert, it is difficult to enrich for nonstable SCVs or to study them in a pure, homogeneous population. Therefore, most studies have used genetically determined stable SCVs to investigate the pathophysiology and clinical significance of SCVs. In these genetically determined stable SCVs, the small colony size is retained upon sub-cultivation. The small colony size of stable SCVs is the consequence of genetic mutations that affect thymidine or hemin biosynthesis, resulting in defects in the electron transport. These SCVs grow slowly, and slow growth confers tolerance to antibiotic[9–11].

Here, we focus on nonstable SCVs, i.e., on colonies whose small size is not a consequence of a mutation and is not retained upon sub-cultivation. Nonstable SCVs and persisters share many important characteristics, including their low frequencies within the population, the ability to revert and an association with chronic infections[11–14]. SCVs and persisters are both frequently found among bacteria recovered from the intracellular milieu or after exposure to stress[15,16]. Here, we provide evidence that SCVs are the product of persisters and arise from bacteria that were dormant within the host. We quantify the frequency of nonstable SCVs in samples from the human and murine host as well as in vitro conditions mimicking host conditions. We determine the bacterial growth dynamics that are responsible for the small size of SCV colonies, and quantify the dynamics of SCV formation and enrichment in vitro and in murine abscesses. Finally, we investigate whether differences in colony size distribution are predictive for antibiotic susceptibility.

## Results

**SCVs are recovered from in vivo and in vitro samples.** Our first aim was to obtain quantitative information about the frequency of nonstable SCVs recovered from patient and mouse abscesses. We first analyzed the distribution of colony sizes of bacteria directly recovered from a *S. aureus* abscess located in the thigh of a patient. To quantify colony size, we determined the colony radius after 24 h of growth. Our quantification showed a distribution of colony sizes with a long tail of smaller colonies, including a small fraction (2.7%) of colonies whose area was five times smaller than the most common colony area, and which therefore qualify as SCV (Fig. 1a; for discussion on the use of colony size ratio for SCV determination, see Supplementary Fig. 4). This contrasted with samples from exponential cultures of the same strain grown in rich nutrient culture medium, which typically did not show any SCVs (under 0.15% of the population; Fig. 1a). Next, we quantified SCVs in a murine abscess model. We observed a distribution of colony sizes similar to the distribution in the sample from the patient, with a larger proportion of small colonies (5.1%, Fig. 1b).

Our next goal was to set up a more controlled in vitro system where we could recapitulate the emergence of SCVs, and study the underlying growth dynamics in more detail. We and others have previously shown that a fraction of *S. aureus* inside a host reside intracellularly inside lysosomes, which are commonly accepted to have a pH of around 5.5[17,18]. In addition, distinct extracellular milieu, such as abscesses, are also acidic (around pH 5.5–7.3)[19–23].

Thus, in the subsequent assays we grew *S. aureus* in liquid cultures at defined acidic and neutral pHs as well as intracellularly in eukaryotic cells. When plating samples from these cultures, we found that low pH induces a shift towards increasing fractions of SCVs, with up to 25% SCV at pH 4 (Fig. 1c, d), in line with previous work[15]. This was also accompanied by a shift in the mean colony size of the largest colonies (Supplementary Fig. 8). In particular, bacteria sampled from a culture grown at pH 5.5 showed a higher fraction of SCVs compared to bacteria recovered from the murine abscess ($P < 0.001$, $t$-test on SCV proportion), but similar to the intracellular model ($P = 0.10$, Fig. 1b). As previously shown[24–26], we found that sub-inhibitory concentrations of gentamicin also induced high numbers of SCVs (see Supplementary Fig. 7).

**Nonstable SCVs are due to late appearance time.** We aimed to analyze why SCV colonies are small, that is, to understand the cellular growth traits that determine differences in colony size. The small colony size of SCVs has previously been attributed to slow growth[9,27,28]. However, in addition to a slow growth rate a small colony size could also be caused by a late initiation of growth or an early cessation of growth compared to larger colonies (Fig. 2a). To discriminate between the three scenarios, we used automated imaging to follow colony growth over time[29,30]. In both the murine abscess model (Fig. 2b) and the in vitro low pH model (Fig. 2c), we observed that the small colonies grew at similar rates to the normal sized colonies ($P = 0.25$, $t$-test on initial growth rate, see Supplementary Fig. 2). When re-plated, small colonies reverted to the colony size distribution observed in samples from exponential cultures (Supplementary Fig. 3). The main difference between small and normal-sized colonies was the time at which they were first detected; we refer to this as "appearance time". As revealed by microscope analysis, late colonies had the same growth rate as

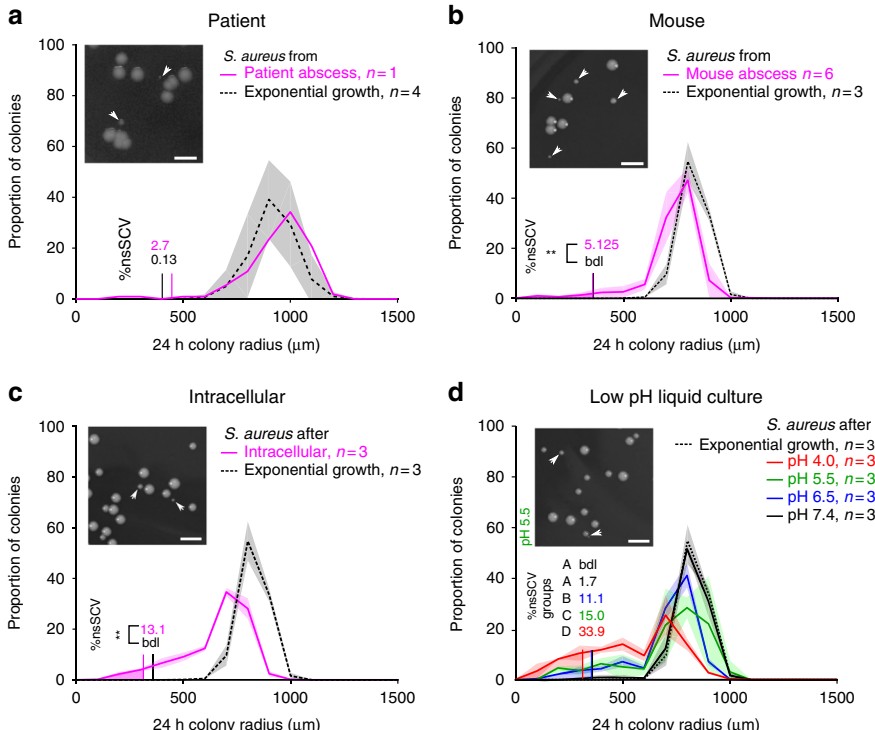

**Fig. 1** *S. aureus* recovered from human and murine abscesses displayed heterogeneous colony sizes including a fraction of small colony variants. Distribution of colony sizes 24 h after plating. Bacteria that were sampled directly from a patient abscess (**a**) or mice abscesses (**b**, see also SI 4), from eukaryotic A549 cells (**c**) and from liquid cultures grown at different pHs (**d**). Bacteria grown exponentially in complex LB medium served as control (dashed black lines). Shaded areas depict the standard deviation. The colored areas under each curve mark colonies whose area was at least five times smaller than the area of the most common colony type, the criterion that is generally used to identify SCVs (the figures depict colony radius instead of colony area, and SCVs would thus be defined as colonies whose radius is at least a factor 2.23, i.e., $\sqrt{5}$, smaller than the radius of the most common colony type. The corresponding frequencies are given above the distributions (labeled as "%nsSCV", "bdl": below detection limit). Insets show representative images of colonies 24 h after plating. Small colonies are indicated by an arrow. All histograms use a binning of 100 μm. Groups in **d** define homogeneous subsets of SCV fractions; treatments with a different group letter. **a**–**d** Show significant differences in the proportion of SCVs (at $P < 0.05$, *t*-test, Bonferroni correction for multiple testing). ** represents a statistical significance of a two-sided *t*-test at 0.01 threshold; shades represent s.d.

early colonies immediately after growth started (Supplementary Fig. 2). This contrasts with growth patterns that we observed in stable SCV strains with mutations that render them auxotrophic; these colonies grew at a slow rate (Fig. 2e) except if the medium was supplemented with hemin to complement the auxotrophy. Together, these results suggest that small colony size in the nonstable SCVs might be the consequence of a later initiation of growth.

**Late colony appearance time reflects long lag times**. While these experiments allowed to analyze growth dynamics once colonies had attained a size of 70 μm (corresponding to about $2 \times 10^6$ bacteria), they did not allow us to follow colony formation from the first cell division on. We thus complemented this macroscopic analysis with time-lapse microscopy to assess whether the small size of SCV colonies is indeed the consequence of a delay in the initiation of the first cell division. Measurements of the first divisional event in the murine abscess and the in vitro models confirmed that the small size of SCV colonies were a consequence of a delayed initiation of cell division of the bacterium that founded the colony (Fig. 2f, g, Supplementary Figs. 1, 2). Importantly, lag-time alone thus explains reversion of nonstable SCVs without requiring a sub-cultivation step. The colony sizes were in direct relation to their appearance times (Supplementary Fig. 2). A few bacteria started to grow as late as 10–20 h (Fig. 2d, g) after the first bacteria had divided. The ratio between colony sizes that is usually used to define SCVs is an approximation, but

not an absolute measure of the phenotypic differences between large and small colonies, since this ratio depends for instance on observation time (see Supplementary Fig. 4). Single-cell lag time is a more robust proxy of the phenotypic differences among bacteria sampled from the different sources analyzed in this study; in the remainder of the manuscript, we will thus refer to SCVs as bacteria with a lag time over 6 h.

**SCVs are formed** de novo **during growth**. Our next goal was to better understand how bacteria with a long lag emerge during growth in vivo and at a low pH. Both conditions lead to an increase in the proportion of SCVs compared to growth in exponential culture at neutral pH. A first possibility is that SCV bacteria emerge de novo in these conditions (in the sense that some bacteria would switch to a phenotypic state associated with a long lag); a second possibility is that SCV bacteria are already present in the inoculum, and that these bacteria are simply enriched because they have higher survival rates in our in vitro and in vivo conditions. One simple test for the first scenario, the de novo emergence, is to ask whether the total number of SCV bacteria after in vivo or in vitro treatment ever exceeds the total number of bacteria that were inoculated into these systems; if it does, then this can only be explained by the emergence of SCV bacteria during in vivo or in vitro growth (possibly in addition to an enrichment through increased survival—the two scenarios are not mutually exclusive).

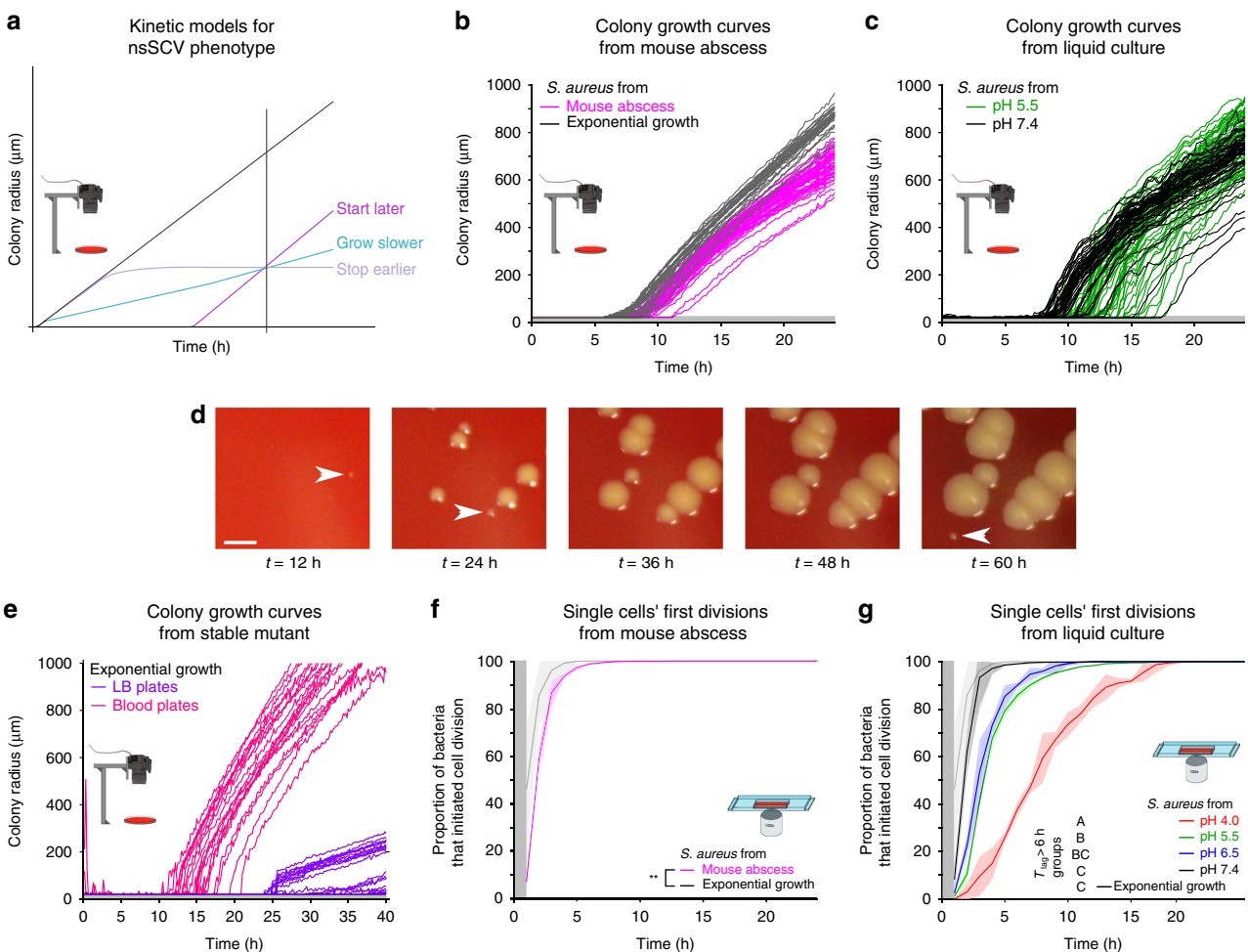

**Fig. 2** SCV formation is a consequence of a late emergence of colonies, and is caused by a delay in the first division at the level of single cells. **a** Schematic of three different scenarios that can explain why a colony is small when measured at a specific time point (vertical black line). **b** Colony growth of bacteria sampled from murine abscesses and compared to bacteria from an exponentially growing culture ($n = 50$ for each condition). Colonies sampled from the murine abscess tended to emerge later than colonies sampled from the exponentially growing culture. **c** Colony growth of bacteria sampled from a liquid culture at neutral (pH 7.4) or acidic (pH 5.5) medium after 3 days of incubation. The same number of colonies is represented for each condition ($n = 50$). Colonies sampled from pH 5.5 tended to emerge later than colonies sampled from pH 7.4. **d** Representative images of colonies acquired at different time points after sampling from a liquid acidic culture medium (pH 4.0). Scale bar, 2 mm. Colonies appeared at different time points (arrows). **e** Colony growth of the stable SCV strain Cowan *hemB::ermB* on LB agar or on LB agar supplemented with sheep blood ($n = 50$ for each condition). The colonies of this stable SCV strain grew at a markedly lower rate on LB plates as compared to blood plates that complement their auxotrophy. **f** Cumulative distributions of the time points of the first cell division of bacteria sampled from a murine abscess model ($n = 5$ mice) and from an exponentially growing culture determined by automated time-lapse microscopy. The vertical gray area marks the period at the beginning of the experiment where cell divisions could occur but not be observed (** for $P < 0.01$). **g** Cumulative distributions of the time points of the first cell division of bacteria sampled from liquid cultures with different pHs and from an exponentially growing culture at neutral pH ($n > 3$). Groups mark homogeneous subsets of the proportion of bacteria with a lag time (time to first division) over 6 h; treatments with a different group letter are significantly different (at $P < 0.05$, *t*-test, Bonferroni correction for multiple testing)

We found that SCVs indeed emerged de novo when culturing bacteria at pH 5.5 in vitro: the absolute number of SCVs observed at the end of the growth phase exceeded the total number of bacteria used for inoculation (Fig. 3a, b). This effect was observed before bacterial cultures reached stationary phase, suggesting that stationary phase was not necessary for the emergence of new SCVs, and that the SCVs we observed in the stationary phase culture could exist before bacteria enter starvation. In fact, we observed that enrichment of SCV bacteria also played a role in increasing their proportion: when we followed our in vitro cultures into stationary phase, we observed that the overall number of bacteria that were able to form a colony decreased over time. However, this decrease was much less pronounced among the SCV bacteria than among the other bacteria (see Fig. 3c, d),

meaning that SCVs were enriched during stationary phase. In the murine abscess model, the total number of SCV colonies obtained from the abscess exceeded the total number of SCVs injected (see Supplementary Fig. 6), again in line with the notion that SCV colonies emerged de novo during the infection. In conclusion, our results suggest that during abscess formation in the murine system, as well as in our in vitro model that mimics growth in the host, bacteria switch to a state where they will form a small colony upon plating. The bacteria in this state will be further enriched during the stationary phase when bacteria die or become unculturable.

**Long lag time confers tolerance to antibiotics.** The appearance of bacteria with a long lag phenotype during the course of an

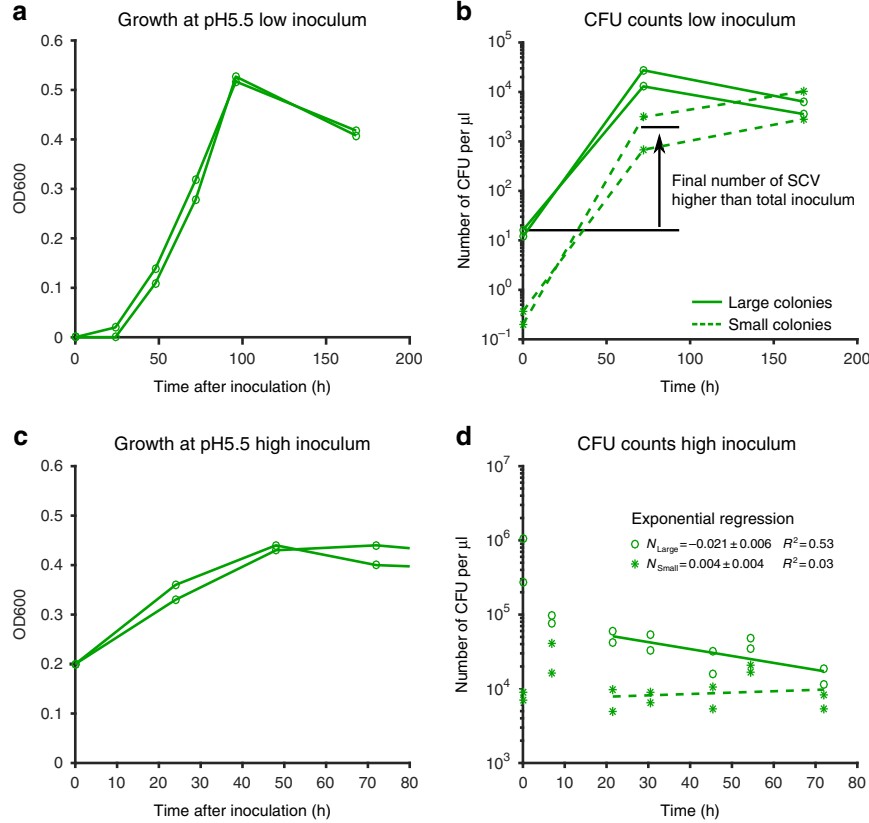

**Fig. 3** SCVs are formed during the growth phase and their proportion increases during stationary phase. **a** When inoculating a low number of bacteria (2 × $10^4$ ml$^{-1}$) in acidic media (pH 5.5), the resulting populations continue to grow for about 100 h. Lines represent changes in optical density over time in two independent replicates. **b** The lines depict the number of SCVs (i.e, colonies with a lag time of more than 6 h; dashed lines, stars) and the number of colonies with larger radius (solid lines, open circles) in these cultures. Seventy-two hours after inoculation, the absolute number of SCVs exceeds the total number of bacteria in the initial inoculum, and SCVs were thus formed during the growth phase. Horizontal black lines are guides to the eye. **c** When bacteria were inoculated at higher concentration, stationary phase was reached earlier and at a similar optical density as in the experiments depicted in **a**. We used these growth experiments for analyzing effects of stationary phase on the frequency of SCVs. **d** After entering stationary phase, the absolute number of SCVs does not vary substantially over time (stars), while the absolute number of colonies with larger radius decreases (empty circles). As a consequence, the proportion of SCVs increases in the population without increase in absolute numbers. Lines are exponential regression for 2 replicates on the indicated time range (dashed: SCV colonies, solid: non-SCV colonies; the estimated rates (for large colonies, L, and small colonies, S) with standard deviations of the slopes are depicted in the panel)

infection prompted us to investigate potential consequences for the effects on antibiotic treatment. Previous studies[31] established that bacteria that are in a state of growth arrest such as stationary phase are tolerant to antibiotics. Most clinically relevant bactericidal antibiotics kill by acting on active targets (e.g., beta-lactams[32]). We thus quantified antibiotic tolerance of the bacteria that only started growing and dividing a few hours after being transferred to new medium (Fig. 4a). We tested whether their numbers would correlate with the number of bacteria surviving exposure to the bactericidal antibiotics flucloxacillin (Fig. 4b) and ciprofloxacin (Fig. 4c) added at ten fold the minimum inhibitory concentration (MIC). Additionally we tested for survival in levofloxacin, another clinically relevant fluoroquinolone, and for concentrations 10-fold and 40-fold the MIC for flucloxacillin, ciprofloxacin and levofloxacin (Supplementary Fig. 5). Bacteria preexposed to acidic pH or sampled from mouse abscesses survived antibiotics significantly better and for longer periods than bacteria preexposed to neutral pH (Fig. 4d). The cumulative time kill curves were similar to the cumulative lag time distributions (compare Fig. 4b, c with f), which is in line with the interpretation that bacteria were tolerant during their lag phase and lost tolerance once they initiated growth. This revealed that after pH 5.5 preexposure, more bacteria survived over an extended period of

time than after pH 7.4 preexposure. For different pH pre-exposures, we observed a correlation between the proportion of bacteria with a lag time over 6 h and the proportion of bacteria that survived a 6 h flucloxacillin exposure (Fig. 4e, $P < 0.001$, rho = 1, Spearman's Rank-Order Correlation). Together, these results are consistent with the scenario that after preexposure to an environment which may be encountered during an infection, such as low pH, a fraction of bacteria with a long lag time tolerate antibiotics better.

**Antibiotics increase the proportion of SCVs.** Our time-lapse microscopy experiments of lag-phase bacteria in the presence of antibiotics revealed that some individual bacterial cells were typically only killed after they had initiated growth and underwent a first cell division (see Supplementary Movie and Supplementary Fig. 10). We quantified the lag times of bacteria that divided at least once in the presence of antibiotics before lysis to ask whether antibiotics had an effect on the lag time distribution of these cells. This analysis did not include bacteria that might have been killed before they initiated growth since we were not able to detect these cases. We observed that lag time increased with the antibiotic concentration. This effect was subtle for

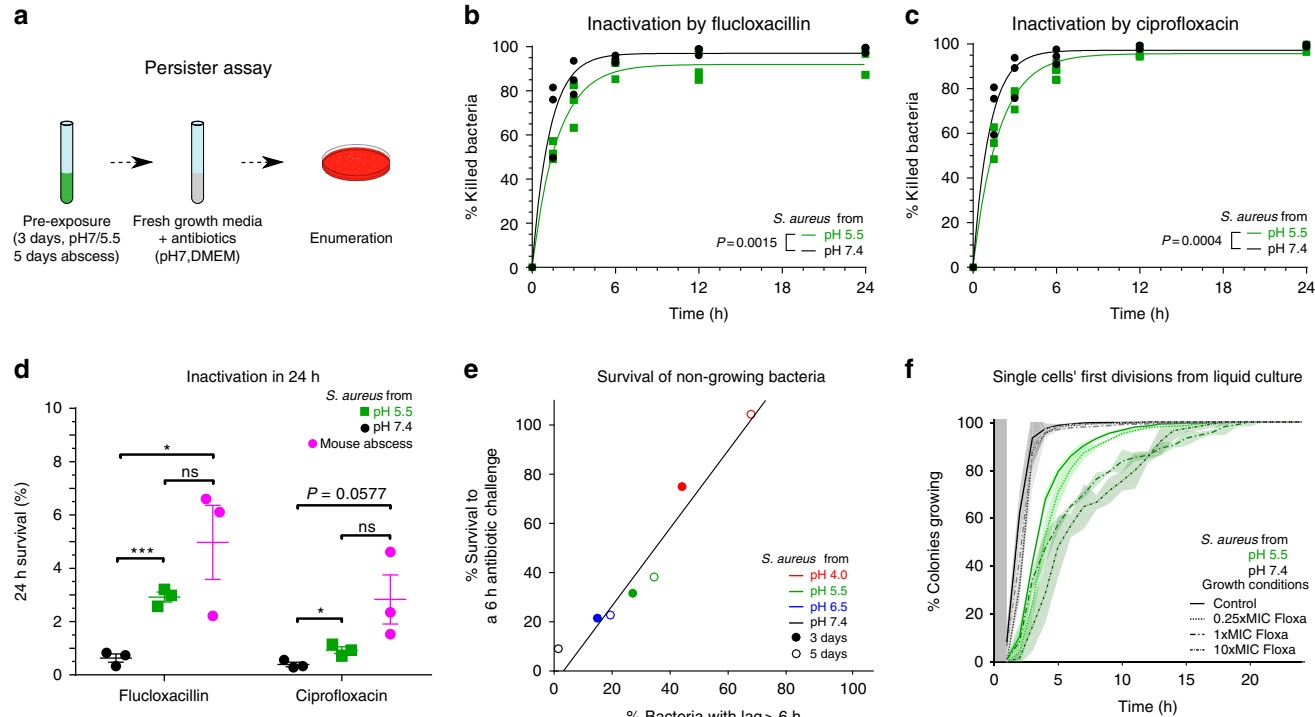

**Fig. 4** The proportion of bacteria in lag phase correlates with the proportion of bacteria surviving antibiotics. **a** Illustration of the persister assay. **b**, **c** Depict the fraction of bacteria that were killed as a function of time upon exposure to the antibiotics flucloxacillin (**b**) and ciprofloxacin (**c**) at 10-fold the minimum inhibitory concentrations. Bacteria were sampled ($n = 3$) from cultures grown in neutral (pH 7.4) or acidic (pH 5.5) medium and then exposed to fresh growth medium containing antibiotics. Each data point was calculated as the ratio between the number of colony forming units (CFU) at a specific time point after antibiotic addition and the number of CFU in the inoculum. Continuous lines represent the one-phase association fit with extra sum of squares $F$-test to compare $K$ values, plateau and $Y_0$ are different. **d** Bacteria recovered from mouse abscesses survived 24 h antibiotic exposure (40-fold MIC) better than bacteria sampled from a stationary-phase pH 7.4 culture. $n = 3$; **** correspond to threshold of $P < 0.05$ and 0.001, respectively for a two-sided student's test. Error bars depict s.e.m. **e** The fraction of bacteria in a sample that had a lag time of more than 6 h upon plating correlated with the fraction of bacteria in the same sample that survived a 6 h exposure to antibiotics. Cultures were sampled at day 3 or 5 to obtain different proportion of bacteria with a lag time of more than 6 h. Black line corresponds to linear fit ($P < 0.001$, rho = 1, Spearman's Rank-Order Correlation). **f** Exposure to antibiotics increased the lag time of individual bacterial cells, as measured by time-lapse microscopy ($n = 3$; $n = 2$ for pH 7.4 at 10-fold MIC). This increase was small for bacteria sampled from cultures with neutral pH (gray curves), and more pronounced for bacteria sampled from acidic cultures (green curves). Curves show averages of 3 replicates, and shaded areas depict s.d.

bacteria preexposed to neutral pH and was more pronounced for bacteria preexposed to low pH (Fig. 4f). This suggests that exposure to antibiotics per se may increase the survival time during which bacteria are tolerant to antibiotics. We observed that the lag time delay was shifted more when the proportion of potentially persisting bacteria was higher due to acidic pH preexposure.

## Discussion

We show that *S. aureus* recovered from abscesses or from in vitro systems mimicking in vivo conditions displayed marked cell-to-cell variation in lag time. This variation in lag time potentially reflects variation in the physiological state of the bacteria recovered from infections or after preexposure to the host environment. More specifically, it is plausible that the bacteria that show a long lag time after sampling were in a state of growth-arrest during the infection. This inference is based on our observation that the number of bacteria that showed a long lag time after sampling correlated closely with the number of bacteria in the sample that were tolerant to antibiotics, as would be expected if these bacteria were growth-arrested, i.e., persisters. This interpretation is in line with previous work showing by direct microscopic analysis that growth arrested *E. coli* can tolerate antibiotics, and non-growing *S. aureus* cultures show high

tolerance when exposed to antibiotics[33]. Persister formation has recently been associated with low intracellular ATP levels[31]. Different factors encountered in the host environment, such as deficiency in nutrients or oxygen[34], most likely further contribute to the low metabolic state of bacteria, in addition to low pH.

We used in vitro acidic conditions or maintaining bacteria inside eukaryotic cells to capture two of the many aspects of the host environment that bacteria may encounter. More specifically, we expect that bacteria are exposed to a range of different conditions in the host, including intracellular microenvironments with low pH and other microenvironments in which bacteria can grow exponentially at a high rate. This could explain differences in the lag time distributions observed between bacteria harvested from hosts and from our in vitro culture model. In our in vitro model, we found that bacteria with a long lag time were continuously produced during growth, and were also enriched during stationary phase. It is important to note that in our model system the bacteria encountered various environmental cues that might trigger entry into a phenotypic state of growth arrest, including acidic pH and increase in bacterial densities. It is thus not possible to establish whether the acquisition of this state was based on an intrinsic mechanism[35,36], or by reduced growth similar to that observed in homogeneously tolerant populations[1]. In both cases, these previously described mechanisms resulted in less efficient killing by bactericidal antibiotics similar to our

observations. One possibility that we did not test here is that bacteria that would produce small colony variants could divide and reproduce during the preexposure period.

Finally, we show that exposure to antibiotics increased the number of bacteria with a long lag time. This emphasizes the importance of carefully evaluating possible side effects of antibiotic treatment. It will be important to investigate whether certain antibiotic treatment regimens may induce tolerance against the antibiotics per se[37], and it will be important to identify regimes that are the most effective in controlling heterogeneous bacterial populations.

## Methods

**S. aureus strains and growth conditions**. *S. aureus* strain Cowan I (ATCC 12598) and the patient isolate (CI1016) were grown on Columbia agar plates plus 5% sheep blood (BioMérieux). The stable SCV mutant strain IIb41 (Cowan *hemB::ermB*)[27] was grown on LB agar plates containing 2.5 mg ml$^{-1}$ erythromycin and supplemented with 5% sheep blood if indicated.

For exponential growth control experiments, overnight culture bacteria were grown in Columbia broth in a 37 °C shaking incubator. Overnight cultures were diluted 1:100, grown for 2 h, and analyzed by automated agar plate imaging after plating (see below).

**Patient sample**. A 54-year-old patient presenting with a subcutaneous *S. aureus* abscess on the left thigh required surgical debridement. The intraoperatively obtained abscess material was used directly for automated agar plate imaging after obtaining informed consent. Experiments were conducted in accordance and in approval with the cantonal ethic commission Zurich.

**Mouse experiment**. *S. aureus* Cowan I was grown to logarithmic phase and 10$^8$ CFUs mixed 1:1 with cytodex beads (Sigma). Bacteria were injected into the flanks of 7–9-weeks-old female C57BL/6 mice (Janvier Laboratory, France)[22]. Mice were sacrificed 5 days post-infection. Abscess pus was harvested, homogenized, and directly analyzed by automated agar plate imaging and single-cell time-lapse microscopy as well as plating serial dilutions for bacterial enumeration.

The protocol ZH251/14 was approved by the institutional animal care and use committee of the University of Zurich and all experiments were conducted in approval of the Cantonal Veterinary Office Zurich.

**Eukaryotic cell infections**. The human lung epithelial carcinoma cell line A549 (ATCC CCL-185) was grown in DMEM 4.5 g ml$^{-1}$ glucose (Life Technologies) supplemented with 10% FBS (GE Healthcare) and L-glutamine (Life Technologies) and infected with *S. aureus* strain Cowan I at a multiplicity of infection of 1[15]. Three hours after infection, the cells were washed with PBS (DPBS; Life Technologies), and flucloxacillin (Actavis; 1 mg ml$^{-1}$ in DMEM) was added to kill any extracellular bacteria. Every day, the washing step was repeated, and fresh medium (containing flucloxacillin 1 mg ml$^{-1}$) was added to the host cells. Supernatants were monitored for the absence of any extracellular bacteria by plating. Three days post-infection, host cells were washed again three times with PBS, lysed (0.08% Triton X-100 from Fluka in DPBS), and analyzed by automated agar plate imaging.

**Growth at low pH**. *S. aureus* were inoculated into the defined pH media (pH 4.0, 5.5, 6.5, and 7.4) at a starting OD$_{600}$ of 0.2[15]. The media consisted of DMEM (4.5 g ml$^{-1}$ glucose, without phenol red, pyruvate, and L-glutamine; Life Technologies) supplemented with 10% FBS and 4 mM of L-glutamine (Life Technologies) and buffers prepared of Na$_2$HPO$_4$ and citric acid (Fluka). pH 7.4 was maintained by 50 mM HEPES (Life Technologies). After incubation at 37 °C in 5% CO$_2$ for three days cultures were analyzed by automated agar plate imaging or single-cell time-lapse microscopy at indicated time points.

**Automated agar plate imaging**. Bacterial dilutions from the different preexposure were plated onto agar plates (Columbia + 5% sheep blood, BioMérieux). Images were acquired every 10 min by triggering Canon EOS 1200D reflex cameras with an Arduino Uno board (Arduino) and optocouplers. The whole setup was placed in a 37 °C incubator.

Colonies were segmented in Matlab 2016a (Mathworks) on the green channel using Sauvola local thresholding[38] and through a two-step circular Hough transform[39]. Images acquired at 24 and 48 h were first automatically processed with this method and then manually curated. Colony radius change over time was obtained automatically by contouring colonies with a local thresholding.

Lag time was inferred assuming a biphasic growth of colonies: bacteria undergoing a short exponential phase with a radial growth rate of 0.41 μm h$^{-1}$ until they reached a diameter of 130 μm, then a linear radial growth rate of 55 μm h$^{-1}$. All parameters were estimated using both microscopic and macroscopic time lapse data.

**Single-cell time-lapse microscopy**. Agar pads were prepared using a custom-made device made of polyvinyl chloride (see Supplementary Fig. 1). Bacterial dilutions were added onto solidified 2% agar (bacteriological grade, BD) with Columbia media (BD) and 5% sheep blood (Life Technologies), then covered with a cover glass. The setup was then placed under microscope at 37 °C at the latest 30 min after inoculation.

Bright field images were acquired using a fully automated Olympus IX81 inverted microscope at 100× resolution (U-FLN-Oil lens) and the Cellsense software. Up to 3000 positions were monitored per experiments and the lag time of 200-1000 bacteria was analyzed. Images were acquired every 30 min starting no later than 30 min after initial inoculation. Time to first division was extracted manually using ImageJ software[40].

**Antibiotic exposure assays**. Bacteria preexposed to acidic conditions or harvested from mouse abscesses were washed and inoculated into DMEM pH 7.4 medium. The DMEM pH 7.4 medium was supplemented with indicated concentrations of flucloxacillin (Actavis) or ciprofloxacin (Bayer). Cultures were incubated at 37 °C in 5% CO$_2$ and culture aliquots were removed at indicated time points to enumerate CFU (surviving bacteria) after washing and plating followed by overnight incubation at 37 °C.

The minimal inhibitory concentration (MIC) of flucloxacillin and ciprofloxacin was assessed by microbroth dilution method. A 0.5 McFarland standard (approximately 10$^8$ cells ml$^{-1}$) was prepared from a fresh *S. aureus* strain Cowan plate and 50 μl of this suspension distributed into the wells of a 96-well plate. Susceptibility testing was performed in either DMEM pH 7.4 or Mueller Hinton Broth and antibiotics incorporated into the medium in serial two-fold dilutions. The 96-well plate was incubated overnight at 37 °C and MIC determined by identifying the well containing the lowest antibiotic concentration that inhibits growth. For *S. aureus* strain Cowan, the MIC was 0.25 mg l$^{-1}$ for flucloxacillin and 1 mg l$^{-1}$ for ciprofloxacin in DMEM pH 7.4, and 0.125 mg l$^{-1}$ for flucloxacillin and ciprofloxacin in Mueller Hinton broth.

**Code availability**. Custom software written for the image analysis of this study is available at GitHub (https://github.com/clementvulin/SCVLongLag).

## Data availability

All the data produced for this study are available from the corresponding author upon request or at figshare (https://figshare.com/articles/RawData_xlsx/6965045).

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

## Acknowledgements

This work was funded by the SNSF grant 310030_146295 and 31003A_176252 (to ASZ), by a CASCADE-FELLOWS PCOFUND-GA-2012-600181 (to CV), by Eawag and ETH Zurich. We are grateful to Kim Lewis (Northeastern University, Boston, USA) for valuable discussion. We thank Roland Regoes, Antoine Frenoy and Nathanaël Hozé (ETH Zurich) for valuable discussions on SCV dynamics, Nathalie Caloz and Alexandra Bernasconi (ETH Zurich) for their help on data analysis and Reto Schüpbach (University Hospital Zurich) for his ideas at the onset of the project.

## Author contributions

C.V., N.L., and M.H. performed in vitro experiments. M.H and A.Z. performed mouse experiments. C.V. and M.H. analyzed data. C.V., N.L., M.H., A.S.Z., M.A. designed the project and wrote the manuscript.

## Additional information

**Competing interests:** The authors declare no competing interests.

