## [Peer Review File · Nature Communications]

Reviewers' comments:

Reviewer #1 (Remarks to the Author):

The manuscript "Prolonged lag time results in small colony variants and reflects a sub-population of persisters in vivo" by Vulin et al. deals with the SCV phenotype of *Staphylococcus aureus* and uses a new definition for SCVs as having a longer lag period than 6h and appear later in growth and are therefore affected in growth with a growth arrest. The authors compare persisters and SCVs and define that the aforementioned SCVs with a long lag phase are a sub-group of persisters.

The authors use new technologies such as automated agar plate reading to follow the colony size of *S. aureus* on agar plates and time-lapse microscopy to follow cell division of single *S. aureus* cells continuously up to 3 division cycles of bacteria grown in neutral medium and in acidic medium, which resemble the in vivo conditions present in the intracellular milieu.

Major comments

1. Although this is an interesting manuscript, which is well written and with sound results, the data, which the authors show, are not entirely new and have been shown in various other studies, to which the authors also refer e.g. Vesga et al. JID 1996; Tuchscherer et al. EMBO Mol Med 2011; Edwards et al. J Bacteriol 2012 to name only a few.

2. A question, which the authors did not touch, is the dynamic of the SCVs and how reversion after sub-culture can be explained.

3. Why did the authors not use aminoglycosides, which have been shown to be very effective in inducing SCVs in vivo (Proctor et al Clin Infect Dis 1995) and in vitro (Balwit et al., J Infect Dis 1994).

4. The effects of β -lactams on growth arrested bacteria are not surprising and are expected. Unexpected is the effect of ciprofloxacin on *S. aureus* tolerance. However, this is not really a drug used for the treatment of *S. aureus*. The authors should use levo- or moxifloxacin instead.

Reviewer #2 (Remarks to the Author):

As a mode of phenotypic heterogeneity, small colony variants (SCVs) have been extensively studied and described in case of *S. aureus*. While SCVs have been reported to occur during normal in vitro growth, they have increasingly been studied in context of chronic infections, and have been implicated in facilitating intracellular survival, by evasion of the host immune system and/or through antibiotic tolerance.

In this study the authors address the growth kinetics of these SCVs and also ask whether these are the same as antibiotic survivors. They quantify the frequency of SCVs in patient and mouse abscesses and as reported in previous studies (References # 5, 10) they find increased frequencies when compared to in vitro cultured bacteria. Also as reported in previous studies (reference 14-18), stresses such as low pH and exposure to intracellular milieu increases the frequency of SCV formation. Subsequently, they setup an automated imaging system to delineate whether the SCV are formed due to slower growth, longer lag or due to earlier arrest of growth. Previous studies addressing these questions largely relied on genetic mutant strains, which were auxotrophic and consequently grew at a slower rate. In the present study, the authors find that SCVs grow at normal rates but have a longer lag period before they initiate growth (longer appearance time). They provide evidence supporting the

notion that SCVs are generated de novo during exponential growth and are enriched during stationary phase. Lastly using flucloxacillin and ciprofloxacin they suggest that the proportion of bacteria surviving antibiotics correlates with proportion of bacteria in lag phase.

Overall, the experiments are planned and executed well and attempts to characterize SCV using single-cell imaging techniques. The manuscript is written well and adds to the recent growing literature highlighting the importance of lag phase in antibiotic persistence. While the growth dynamics data is convincing, the data on antibiotic survivors being lag phase cells is largely inferential from correlation data and should be toned down a bit. Also, it would be useful to use bacteria obtained from abscesses rather than only low pH-exposed bacteria to draw their conclusions (see comment 1 below).

Specific comments:

- While the authors quantify and demonstrate SCVs from patient and mouse abscesses in Fig. 1, most of the subsequent figures (Fig. 2-4) use only SCVs generated from low pH liquid culture. If one looks at the distribution of colony sizes (focusing only on the non-SCV fraction in Fig. 1A-D), the colonies from patient and mouse abscesses are as large as the ones from exponential growth, whereas the colonies from intracellular and especially from low pH growth are particularly smaller. This suggests that these environments are not only enriching for SCVs but are potentially deficient in nutrients or are reducing the metabolic growth potential through stress. Since the subsequent experiments – especially on antibiotics survivors are only done using low pH conditions, this makes it more difficult to directly implicate only SCVs. This is all the more relevant since one of the criticisms the authors posit against previous studies on SCVs is that those studies are flawed because the SCV mutant strains grow at a slower rate. In this regard, it would be useful to provide a table of mean colony sizes of SCV and non-SCV populations in different conditions.

- The authors claim, from data shown in Figure 2, that the SCVs grow with the same rates as normal colonies (2B, C) but become SCV due to longer appearance time (2F and G). However, the data shown in Fig. 2F shows no difference in the proportion of cells showing the first division event, between exponential growth cells (SCVs below detection limit) and cells from mouse abscesses (SCV ~5%). As in previous comment, the differences seen in Fig. 2G could be result of the extremely stressful pH conditions, which render the slower growth of the whole population.

- Figure 3. While the experiments shown do suggest that SCVs are generated de novo, it does not clearly demonstrate that these SCVs are formed due to a phenotypic switch from normal colony variants. It can be envisaged that the SCVs are stably maintained and are propagating as SCVs, which would also result in the distribution shown in Fig. 3B. This would suggest a possibility that the normal and SCV populations co-exist without interchanging between themselves. Maybe the authors can suggest this as alternate possibility or provide data to support switching.

- Fig. 4. Why were different fold MICs used for the two antibiotics (2.5X for Cipro and 10X for Fluco).

- Are the growth rates of *S. aureus* growing at pH 7.4 and pH 5.5 different? If so, then that would be cause for decreased killing at pH 5.5, not related with SCVs. Also is the activity of the drugs equally maintained at pH 5.5 and pH 7.4?

- Are the curves shown in Fig. 4A and 4B statistically different?

- The data shown in Fig. 4D is not entirely clear. The authors claim that individual cells were killed only after they had initiated growth and underwent a first cell division. Is this true for all lysis events plotted or only for the cell shown in Supplementary video 1? If true for all lysis events, would a similar

pattern be maintained even for much higher concentrations of drug or would cells lyse without initiating growth.

Point by point reply

Reviewer #1 (Remarks to the Author):

The manuscript "Prolonged lag time results in small colony variants and reflects a sub-population of persisters in vivo" by Vulin et al. deals with the SCV phenotype of Staphylococcus aureus and uses a new definition for SCVs as having a longer lag period than 6h and appear later in growth and are therefore affected in growth with a growth arrest. The authors compare persisters and SCVs and define that the aforementioned SCVs with a long lag phase are a sub-group of persisters. The authors use new technologies such as automated agar plate reading to follow the colony size of S. aureus on agar plates and time-lapse microscopy to follow cell division of single S. aureus cells continuously up to 3 division cycles of bacteria grown in neutral medium and in acidic medium, which resemble the in vivo conditions present in the intracellular milieu.

Thank you for your positive and constructive comments.

Major comments

1. Although this is an interesting manuscript, which is well written and with sound results, the data, which the authors show, are not entirely new and have been shown in various other studies, to which the authors also refer e.g. Vesga et al. JID 1996; Tuscherr et al. EMBO Mol Med 2011; Edwards et al. J Bacteriol 2012 to name only a few.

We thank the reviewer for this important comment. In the revised manuscript, we have incorporated a description of the differences between our data and the current knowledge of SCVs dynamics to emphasize the novelty of our work. Moreover, we have added a comment in the introduction highlighting the usefulness of colony time-lapse and single cell microscopy to shed light on bacterial persistence. Our findings emphasize that lag time, not growth rate is likely responsible for SCV phenotypes observed in clinical samples and previous studies.

- We modified the following paragraph accordingly (additional text in gray):

“Most SCVs recovered in vitro and from murine infection models revert to a normal sized colony upon sub-cultivation and are consequently termed nonstable SCVs (12). Within a population, these nonstable SCVs usually occur at a low frequency, which increases after exposure to the host intracellular milieu or after exposure to distinct stress conditions, such as acidic pH (12–18) (for *Salmonella enterica*, see (19)). Due to their low frequency within a population, detection and subsequent characterization of nonstable SCVs is challenging and their growth dynamics have not been studied so far. Furthermore, due to their propensity to revert, it is difficult to enrich for nonstable SCVs or to study them in a pure, homogeneous population. Therefore, most studies have used genetically determined stable SCVs to investigate the pathophysiology and clinical significance of SCVs (9). In these genetically determined stable SCVs, the small colony size is retained upon sub-cultivation. The small colony size of stable SCVs is the consequence of genetic mutations that affect thymidine or hemin biosynthesis, resulting in defects in the electron transport. These SCVs grow slowly, and slow growth confers tolerance to antibiotic (Vesga, 1996).

2. A question, which the authors did not touch, is the dynamic of the SCVs and how reversion after sub-culture can be explained.

We appreciate the reviewers' comment. We have further developed this aspect in the manuscript by adding new data, which we have included as supplementary material:

- Following up on the reviewers comment we have added data showing that when we transferred a small colony at very early stage to a new agar plate we could recover a normal phenotype (in terms of colony size), showing that the phenotype was lost at very early stage of colony growth. This data is presented in a new supplementary figure 3 and mentioned in text as follows:
“...sized colonies ($p=0.25$, t-test on initial growth rate, see **SI 2**). When re-plated, small colonies reverted to the colony size distribution observed in samples from exponential cultures (SI3). The main difference...”
- In the main text, we emphasized our observation (based on microscopy) that colony growth rate of late and early colonies reached the same exponential rate:
“...we refer to this as ‘appearance time’ (see **SI 3**). As revealed by microscope analysis, late colonies had the same growth rate as early colonies immediately after growth started (SI3). This contrasts with...”
- We now emphasize that the lag time alone explains why reversions are observed. In the manuscript we added the following section (in grey):
“of a delayed initiation of cell division of the bacterium that founded the colony (**Fig. 2F and 2G, SI 2**). Importantly, lag-time alone thus explains reversion of nonstable SCVs without requiring a sub-cultivation step. The colony size was in direct relation to their appearance time (SI 2). A few bacteria started to grow as late as 10 to 20 hours”

3. Why did the authors not use aminoglycosides, which have been shown to be very effective in inducing SCVs *in vivo* (Proctor et al Clin Infect Dis 1995) and *in vitro* (Balwit et al., J Infect Dis 1994)?

Aminoglycosides are indeed often used to produce SCVs in the lab. We initially did not use aminoglycosides because we aimed to assess the effect of the host environment (or acidic pH) on SCV formation. However, following the reviewer's suggestion we now performed additional experiments in which we analyzed the induction of SCVs by aminoglycosides. This new data is presented in Supplementary material 7.

- We mention these results in the text as follows:
“In particular, bacteria sampled from a culture grown at pH 5.5 showed a higher fraction of SCVs as compared to bacteria recovered from the murine abscess ($p<0.001$, t-test on SCV proportion), but similar to the intracellular model ($p=0.10$, **Fig. 1B**). As previously shown (Balwit, 1994; von Eiff, 1997; Musher, 1977), we found that sub-inhibitory concentrations of gentamicin also induced high numbers of SCVs (see Supplementary material 7).

4. The effects of β -lactams on growth-arrested bacteria are not surprising and are expected. Unexpected is the effect of ciprofloxacin on *S. aureus* tolerance. However, this is not really a drug used for the treatment of *S. aureus*. The authors should use levo- or moxifloxacin instead.

According to the reviewer's suggestion we have expanded our experiments and performed additional time-kill curves with levofloxacin reflecting the ciprofloxacin data. We have included this additional information as new supplemental figure 5.

“...bactericidal antibiotics flucloxacillin (Fig. 4B) and ciprofloxacin (Fig. 4C) added at 10 times the minimum inhibitory concentrations (MIC). Additionally we tested for survival in levofloxacin, another clinically relevant fluoroquinolone, and for concentrations 10 and 40 fold above MIC for flucloxacillin, ciprofloxacin and levofloxacin (SI 5). Bacteria preexposed to acidic pH...”

We also modified the following paragraph:

“... Previous studies (32) established that bacteria that are in a state of growth arrest such as stationary phase are tolerant to antibiotics. Most clinically relevant bactericidal antibiotics kill by acting on active targets (e.g. beta-lactams, Tuomanen, 1986). We thus quantified antibiotic tolerance...”

Reviewer #2 (Remarks to the Author):

*As a mode of phenotypic heterogeneity, small colony variants (SCVs) have been extensively studied and described in case of *S. aureus*. While SCVs have been reported to occur during normal in vitro growth, they have increasingly been studied in context of chronic infections, and have been implicated in facilitating intracellular survival, by evasion of the host immune system and/or through antibiotic tolerance. In this study the authors address the growth kinetics of these SCVs and also ask whether these are the same as antibiotic survivors. They quantify the frequency of SCVs in patient and mouse abscesses and as reported in previous studies (References # 5, 10) they find increased frequencies when compared to in vitro cultured bacteria. Also as reported in previous studies (reference 14-18), stresses such as low pH and exposure to intracellular milieu increases the frequency of SCV formation. Subsequently, they setup an automated imaging system to delineate whether the SCV are formed due to slower growth, longer lag or due to earlier arrest of growth. Previous studies addressing these questions largely relied on genetic mutant strains, which were auxotrophic and consequently grew at a slower rate. In the present study, the authors find that SCVs grow at normal rates but have a longer lag period before they initiate growth (longer appearance time). They provide evidence supporting the notion that SCVs are generated de novo during exponential growth and are enriched during stationary phase. Lastly using flucloxacillin and ciprofloxacin they suggest that the proportion of bacteria surviving antibiotics correlates with proportion of bacteria in lag phase.*

Overall, the experiments are planned and executed well and attempts to characterize SCV using single-cell imaging techniques. The manuscript is written well and adds to the recent growing literature highlighting the importance of lag phase in antibiotic persistence.

We thank the reviewer for the positive comments and appreciate the suggestions for further improvements, which we have addressed below.

While the growth dynamics data is convincing, the data on antibiotic survivors being lag phase cells is largely inferential from correlation data and should be toned down a bit.

Following the reviewers comments we have toned down the phrasing regarding the link between antibiotic survivors and lag phase cells. We also added a more detailed discussion of previous work that addresses this issue:

“This variation in lag time potentially reflects variation in the physiological state of the bacteria recovered from both infections or after pre-exposure to the host environment. More specifically, it is plausible that the bacteria that show a long lag time after sampling were in a state of growth-arrest during the infection. This inference is based on our observation that the

number of bacteria that showed a long lag time after sampling correlated closely with the number of bacteria in the sample that were tolerant to antibiotics, as would be expected if these bacteria were growth-arrested, i.e. persisters. This interpretation is in line with previous work showing by direct microscopic analysis that growth arrested *E. coli* can tolerate antibiotics (Balaban, 2004), and non-growing *S. aureus* cultures show high tolerance when exposed to antibiotics (Conlon, 2016). Persister formation has recently been associated with low intracellular ATP levels (Conlon, 2016). Different factors encountered in the host environment, such as deficiency in nutrients or oxygen (Fang, 2006), most likely further contribute to the low metabolic state of bacteria, in addition to low pH.

We used *in vitro* acidic conditions or maintaining bacteria inside eukaryotic cells to capture two of the many aspects of the host environment. More specifically, we expect that bacteria are exposed to a range of different conditions in the host, including intracellular microenvironments with low pH and other microenvironments in which bacteria can grow exponentially at a high rate. This could explain differences in the lag time distributions observed between bacteria harvested from host and from our *in vitro* culture model.

Also, it would be useful to use bacteria obtained from abscesses rather than only low pH-exposed bacteria to draw their conclusions (see comment 1 below).

We agree with the reviewer and have included data on bacteria retrieved from mouse abscesses and compared 24 h persister data from mouse abscess-derived bacteria to bacteria that were exposed to pH 5.5 or pH 7.4. The results corroborate that abscesses contain higher numbers of bacterial persisters. These additional data have been included in Figure 4D.

Specific comments:

- While the authors quantify and demonstrate SCVs from patient and mouse abscesses in Fig.1, most of the subsequent figures (Fig. 2-4) use only SCVs generated from low pH liquid culture. If one looks at the distribution of colony sizes (focusing only on the non-SCV fraction in Fig. 1A-D), the colonies from patient and mouse abscesses are as large as the ones from exponential growth, whereas the colonies from intracellular and especially from low pH growth are particularly smaller. This suggests that these environments are not only enriching for SCVs but are potentially deficient in nutrients or are reducing the metabolic growth potential through stress. Since the subsequent experiments – especially on antibiotics survivors are only done using low pH conditions, this makes it more difficult to directly implicate only SCVs. This is all the more relevant since one of the criticisms the authors posit against previous studies on SCVs is that those studies are flawed because the SCV mutant strains grow at a slower rate. In this regard, it would be useful to provide a table of mean colony sizes of SCV and non-SCV populations in different conditions.

We thank the reviewer for this important comment. As proposed by the reviewer, we added a table of mean colony sizes of SCV and non-SCV populations in different conditions (supplementary material 8). We agree that nutrient limitations, in combination with pH stress, could explain the delay until the first bacteria grow. We also agree that more detailed experiments, allowing to pinpoint subtle changes in colony size, will open up additional biological questions such as the one proposed by the reviewer.

We now mention that ATP levels (linked with both pH stress and starvation) are linked to the persister phenotype, and could possibly explain our observations on non-stable SCV:

“...non-growing *S. aureus* cultures show high tolerance when exposed to antibiotics (Conlon, 2016). Persister formation has recently been associated with low intracellular ATP levels (Conlon, 2016). Different factors encountered in the host environment, such as deficiency in nutrients or oxygen, most likely further contribute to the low metabolic state of bacteria, in addition to low pH.”

The new table is now mentioned in the text as follows:

“...we found that low pH induces a shift towards increasing fractions of SCVs, with up to 25% SCV at pH 4 (Fig. 1C, 1D), in line with previous work (14). This was also accompanied by a shift in the mean colony size of the largest colonies (SI 8). In particular, bacteria sampled from a culture grown at pH 5.5... “

Please note that no delay for the early growing bacteria (resulting in larger colonies) was observed at pH 7.4 after 3 days of growth, even long after saturation was reached. We do believe that, as the reviewer suggests, starvation can be important in the formation of SCVs in combination with low pH. However, we believe that starvation is not the only reason for the formation of SCVs, as suggested by the appearance of *de novo* SCVs during exponential growth (figure 3). Following the reviewer's comment, we now further emphasized this point in the main text:

“... This effect was observed before bacterial cultures reached stationary phase, suggesting that stationary phase was not necessary for the emergence of new SCVs, and that the SCVs we observed in the stationary phase culture could exist before bacteria enter starvation. In fact, we observed that enrichment of SCV bacteria also played a role in increasing their proportion: when we followed our *in vitro* cultures into stationary phase, we observed that the overall number of bacteria that were able to form a colony decreased over time. However...”

- *The authors claim, from data shown in Figure 2, that the SCVs grow with the same rates as normal colonies (2B, C) but become SCV due to longer appearance time (2F and G). However, the data shown in Fig.2F shows no difference in the proportion of cells showing the first division event, between exponential growth cells (SCVs below detection limit) and cells from mouse abscesses (SCV ~5%). As in previous comment, the differences seen in Fig. 2G could be result of the extremely stressful pH conditions, which render the slower growth of the whole population.*

We thank the reviewer for this important comment. We agree. The bacteria sampled from abscesses were exposed to a number of external factors that could influence the time at which they initiate cell division after plating. We think that one reason for the somewhat lower SCV counts observed in the abscess samples is that only a fraction of the bacteria in such a sample were exposed to intracellular conditions and low pH; another fraction of the bacteria in an abscess might experience exponential growth. This could explain why there was no statistically significant effect in the dataset that we presented in the previous version of the manuscript. We now use data from 5 mice, and have incorporated the data into Figure 2F. The difference observed between the proportions of bacteria appearing after 6h in mice versus stationary phase bacteria in the new dataset is statistically significant. We have now added an additional comment in the discussion:

“We used *in vitro* acidic conditions or maintaining bacteria inside eukaryotic cells to capture one of the many aspects of the host environment that bacteria may encounter. More specifically, we expect that bacteria are exposed to a range of different conditions in the host, including intracellular microenvironments with low pH and other microenvironments in which bacteria can grow exponentially at a high rate. This could explain differences in the lag time distributions observed between bacteria sampled from hosts and from our *in vitro* culture model.”

- Figure 3. While the experiments shown do suggest that SCVs are generated *de novo*, it does not clearly demonstrate that these SCVs are formed due to a phenotypic switch from normal colony variants. It can be envisaged that the SCVs are stably maintained and are propagating as SCVs, which would also result in the distribution shown in Fig.3B. This would suggest a possibility that the normal and SCV populations co-exist without interchanging between themselves. Maybe the authors can suggest this as alternate possibility or provide data to support switching.

We agree with this possibility and added a corresponding sentence to the revised manuscript:

“...our observations. One possibility that we did not test here is that bacteria that would produce small colony variants could divide and reproduce during the pre-exposure period.”

- Fig. 4. Why were different fold MICs used for the two antibiotics (2.5X for Cipro and 10X for Fluco).

We now have redone the experiments using 10 fold and 40 fold MIC. The additional data is included in the modified figure 4 and supplementary figure 5; the new data supports the previous conclusions.

- Are the growth rates of *S. aureus* growing at pH 7.4 and pH 5.5 different? If so, then that would be cause for decreased killing at pH 5.5, not related with SCVs. Also is the activity of the drugs equally maintained at pH 5.5 and pH 7.4?

The growth rates at pH 7.4 and 5.5 are indeed different. In order to address this point, we have added this information to the supplementary material 9.

However, we would like to point out that after pre-exposure to different pH values, bacteria are harvested and the persister assay is performed under identical growth conditions (i.e. always at pH 7.4). Thus, the antibiotic activity and stability is identical. The reviewer’s comment made us realize that the description of our experimental design was not sufficiently clear. We have included a schematic drawing of our persister assay (Figure 4A).

- Are the curves shown in Fig. 4A and 4B statistically different?

We now have performed statistical tests; the curves are statistically different.

- The data shown in Fig. 4D is not entirely clear. The authors claim that individual cells were killed only after they had initiated growth and underwent a first cell division. Is this true for all lysis events plotted or only for the cell shown in Supplementary video 1? If true for all lysis events, would a similar pattern be maintained even for much higher concentrations of drug or would cells lyse without initiating growth.

We thank the reviewer for this comment. Some bacteria were indeed killed before they initiate their first division, and were not represented in figure 4D, which represents only first division events. We now clarify this in the text:

“Our time-lapse microscopy experiments of lag-phase bacteria in the presence of antibiotics revealed that some bacterial cells were only killed after they had initiated growth and underwent a first cell division (see supplementary video 1 and SI10). We quantified the lag times of bacteria that divided at least once in the presence of antibiotics before lysis to ask whether antibiotics had an effect on the lag time distribution of these cells. This analysis did not include bacteria that might have been killed before they initiated growth since we were not able to detect these cases.”

Following the reviewer’s suggestion, we performed an experiment using 10 times MIC antibiotic concentration in addition. At this antibiotic concentration, we observed that few cells completed one division and we never observed two divisions. We added this additional information to figure 4 and to supplementary material 10 as well as in supplementary video 1.

Also, when measuring the MIC of flucloxacillin in our laboratory again, the concentrations that we previously used corresponded to 0.25 and 1 times MIC, not 0.5 and 2 times MIC as initially reported. We have adjusted this in the new version of the manuscript. After adding the new experiments performed at 10 times MIC, our dataset now covers a range of concentrations from substantially below to substantially above MIC.

REVIEWERS' COMMENTS:

Reviewer #1 (Remarks to the Author):

no further comments

Reviewer #2 (Remarks to the Author):

The authors have addressed my comments to the previous version of the manuscript and have included additional data to support their conclusions.

I'm satisfied with the current version of the manuscript.

I found two small typos which the authors should correct.

- Pg 2. Abstract Line 4- SCVs is mentioned as SVCs.
- Pg 14, Ln 11 Fluoroquinolone is misspelt